# Role of Yes-Associated Protein in Psoriasis and Skin Tumor Pathogenesis

**DOI:** 10.3390/jpm12060978

**Published:** 2022-06-16

**Authors:** Jinjing Jia, Yuqian Wang, Xiumei Mo, Dacan Chen

**Affiliations:** 1State Key Laboratory of Dampness Syndrome of Chinese Medicine, the Second Affiliated Hospital of Guangzhou University of Chinese Medicine, Guangzhou 510120, China; 13310980167@gzucm.edu.cn (J.J.); szyadteam@gzucm.edu.cn (X.M.); 2Department of Dermatology, the Second Affiliated Hospital of Guangzhou University of Chinese Medicine, Guangzhou 510120, China; 3Guangdong-Hong Kong-Macau Joint Lab on Chinese Medicine and Immune Disease Research, Guangzhou University of Chinese Medicine, Guangzhou 510120, China; 4Department of Dermatology, the Second Affiliated Hospital of Xi’an Jiaotong University, Xi’an 710049, China; wyq0812@stu.xjtu.edu.cn

**Keywords:** Yes-associated protein, Hippo signaling pathway, psoriasis, skin tumors, cutaneous squamous cell carcinoma, melanoma

## Abstract

Psoriasis and skin tumors (such as basal cell carcinoma, squamous cell carcinoma, and melanoma) are chronic diseases that endanger physical and mental health, and yet the causes are largely unknown and treatment options limited. The development of targeted drugs requires a better understanding of the exact pathogenesis of these diseases, and Yes-associated protein (YAP), a member of the Hippo signaling pathway, is believed to play an important role. Psoriasis and skin tumors are characterized by excessive cell proliferation, abnormal differentiation, vasodilation, and proliferation. Here, we review the literature related to YAP-associated disease mechanisms and discuss the latest research. YAP regulates cell apoptosis, proliferation, and differentiation; inhibits cell density and intercellular contacts and angiogenesis; and maintains the three-dimensional structure of the skin. These mechanisms may be associated with the occurrence and development of psoriasis and skin tumors. The results of recent studies have shown that YAP expression is increased in psoriasis and skin tumors. High expression of YAP in psoriasis and skin tumors may indicate its positive functions in skin inflammation and malignancies and may play an important role in disease pathogenesis. The study of new drugs targeting YAP can provide novel approaches for the treatment of skin diseases.

## 1. Introduction

Psoriasis is a chronic inflammatory disease of the skin, with an incidence rate of approximately 0.1–4% and affecting all ages. Given its high incidence, recurrence is common. Lesions are often located on exposed parts of the body, which results in significant psychological and social pressure on patients and seriously affects their physical and mental health and quality of life [1,2,3]. Patients with psoriasis are also at increased risk of metabolic syndrome [4,5,6]. However, the etiology and pathogenesis of psoriasis are not completely known. There is a general consensus that adverse external stimuli, such as infection, mental stress, trauma, and medications, result in abnormalities in metabolism and regulation of the immune system. Inflammatory cell chemotaxis, infiltration, and the release of inflammatory mediators under the joint action of internal factors, external factors (acting on certain genotypes), and other risk factors lead to pathological changes, such as excessive proliferation and abnormal differentiation of epidermal keratinocytes [7]. Although biological agents have achieved good results in treating psoriasis in recent years, significant limitations remain, such as high cost, inducibility of infectious diseases and tumors, and recurrence after medication withdrawal. Indeed, psoriasis often requires long-term or even lifelong maintenance treatment [8].

Skin tumors, such as melanoma and non-melanoma skin cancers (NMSCs), are cell proliferative diseases of the intradermal or subcutaneous tissues [9]. The most common types of NMSCs are basal cell carcinoma (BCC) and cutaneous squamous cell carcinoma (cSCC), which often occur at exposed sites. Approximately 5.4 million cases of BCC and cSCC are diagnosed annually [10]. Because the malignancy of these cancers is low and most of them can be surgically removed, the prognosis is relatively good. However, because these tumors mainly occur in the head and face, they can damage the local skin and surrounding tissues, resulting in disfigurement; at later stages, they can have serious consequences due to the invasion of bones and nerves, consequently affecting the daily life of patients and causing serious harm to the physiological and mental health of patients [11]. The incidence rate of cutaneous melanoma is not high, accounting for only 4% of malignant tumors; however, its degree of malignancy is high, metastasis occurs early, and the prognosis is generally very poor. The mortality rate accounts for 80% of all skin tumors [12].

Currently, the etiology of psoriasis and skin tumors remains unclear. Ultraviolet radiation, chemical carcinogens, and viral infection may be related to the pathogenesis of skin tumors [11], but they have not been fully clarified. There is an urgent need to uncover the exact mechanisms of pathogenesis in order to develop more effective and targeted therapeutic drugs. Accordingly, we carried out an extensive review of the literature, including articles in the PubMed database.

## 2. Composition of the YAP and Hippo Signaling Pathways

Yes-associated protein (YAP) is a key member of the Hippo signaling pathway. It was originally identified in *Drosophila*, with a molecular weight of 65 kDa, and is also known as YAP65 [13]. YAP was confirmed to be encoded by the *yap* gene on human chromosome 11q22. It is a proline-rich protein that plays a role in controlling the cell number and organ size in *Drosophila* development by coordinating cell proliferation, the cell cycle, and apoptosis [14,15]. The Hippo signaling pathway includes three main components: upstream kinase cascade initiation, kinase cascade core, and transcription-related activity. The core components of the mammalian kinase cascade include Mst1/2, Sav1, LATS1/2, Mob1, YAP, and TAZ (transcriptional coactivator with the PDZ-binding motif). Dong et al. [16] first determined the signal transmission sequence of the Hippo–YAP pathway in mammals; when an external stimulation signal activates the Hippo pathway, Mst1/2 and Sav form a complex, which binds to LATS1/2 through the adaptor protein WW45, resulting in activation of LATS1/2, which subsequently phosphorylates YAP at the ser127 site. YAP transcriptional activity depends on its subcellular localization: YAP is active in its dephosphorylated state and can thus enter the nucleus and activate gene transcription by interacting with transcription factors TEAD (TEA domain family members) 1–4, Smads, p63/p73, Runx, and ErbB4. When phosphorylated through the upstream kinase cascade, YAP binds to the cytoplasmic 14-3-3 protein, is inactivated, and remains in the cytoplasm, where it cannot act as a transcription factor (Figure 1).

## 3. Possible Mechanisms of YAP in Psoriasis and Skin Tumor Pathogenesis

YAP and its upstream pathway (the Hippo signaling pathway) play important roles in regulating cell proliferation and apoptosis to regulate the growth, development, and size of tissues and organs; epithelial mesenchymal transformation (EMT); intercellular contact inhibition; and stem cell self-renewal. Tumor research accounts for the vast majority of studies on YAP [17,18]. YAP is highly expressed in lung cancer [19], breast cancer [20], ovarian cancer [21,22], colon cancer [23], liver cancer [24], and other cancers. In some tumors, YAP expression is positively correlated with the survival rate and degree of tumor malignancy. Psoriasis and tumors are characterized by excessive cell proliferation, abnormal differentiation, vasodilation, and hyperplasia. Therefore, anything that can affect these pathophysiological mechanisms may be a key link in their pathogenesis.

### 3.1. YAP Regulates Apoptosis

The importance of the Hippo–YAP signaling pathway in regulating apoptosis has been confirmed by a number of studies. In *Drosophila*, the Yorkie (Yki) protein, a homologue of YAP, can inhibit the activity of the *Drosophila* apoptosis-related gene *reaper* (RPR) by promoting the binding of p53 to ASPP1 (the apoptosis-stimulating protein of p53). Yki can also inhibit the activation of defective head involvement (HID), both of which inhibit *Drosophila* apoptosis [25]. The Hippo–YAP signaling pathway can also increase the expression of *Drosophila* inhibitor of apoptosis protein 1 (DIP1) [26]. YAP also inhibits apoptosis in human periodontal ligament stem cells [27], human liver cancer cells [28], endometrial stromal cells [29], and meningioma cells [30].

YAP can also promote apoptosis in some cases. During DNA damage, the promyelocytic leukemia gene (PML) recruits YAP and p73 into the nucleus. YAP acts as a transcriptional activator to enhance binding with p73 and activates the transcription of apoptosis-related genes [31]. YAP accelerates amyloid-β-peptide (Aβ)-induced apoptosis through nuclear translocation, induces p73-mediated Bax expression and activation, and promotes the activation of the apoptosis-related protein caspase-3 [32]. In addition, YAP mediates c-Jun-dependent apoptosis [33].

### 3.2. YAP Regulates Cell Proliferation and Differentiation and Maintains the Three-Dimensional Structure of Skin

In mice and *Drosophila*, YAP overexpression induces the proliferation of undifferentiated intestinal progenitor cells. Reduced YAP expression induces the differentiation of intestinal progenitor cells. YAP activation or inhibition of upstream negative regulatory proteins can induce the activation or enhancement of the Notch and Wnt signaling pathways, which are related to the inhibition of stem cell differentiation and promotion of stem cell proliferation [34,35]. Schlegelmilch et al. [36] confirmed that YAP regulates epidermal stem cell proliferation and maintains the three-dimensional structure of the skin by interacting with the transcription factor TEAD; the epidermis of YAP-knockout mice became thinner, the stratum corneum decreased, and the arrangement of the epidermal structure became disordered. Zhang et al. [37] also reported high expression of YAP in monolayer basal epidermal progenitor cells, and that the expression of nuclear YAP decreased gradually with age, which was related to the proliferation potential of epidermal progenitor cells. YAP promotes the proliferation of basal epidermal progenitor cells and inhibits terminal differentiation. In vitro studies have shown that, when YAP is activated, the proliferation rate of primary mouse keratinocytes (MKSs) increases, whereas the rates of differentiation and apoptosis decrease, and these characteristics are reversed after inhibiting YAP expression. YAP acts as a molecular switch for epidermal stem/progenitor cell activation. The C-terminus of YAP was found to regulate the balance between stem/progenitor cell proliferation and differentiation in the postnatal interfollicular epidermis [38]. Overexpression of YAP can promote immortalized proliferation of human primary keratinocytes, hinder their normal differentiation process, increase the expression of the epithelial proliferation markers p63 and PCNA, and decrease the expression of the epidermal differentiation markers 14-3-3σ and LEKTI [39]. YAP knockout resulted in a reduced level of expression of transforming growth factor (TGF)-β, a decrease in the proliferation rate of epidermal basal cells, and hindrance of skin wound healing. Interestingly, skin wound healing is dependent on YAP expression [40]. YAP is mainly localized in the cytoplasm of differentiated cells [41]; however, when epithelial tumor cells lose polarity and exhibit enhanced invasiveness, YAP is mainly located in the nucleus. Therefore, the regulation of YAP subcellular localization is crucial for the conversion between proliferation and terminal differentiation [36].

### 3.3. YAP Regulates Cell Density and Intercellular Contact Inhibition

Contact inhibition refers to the biological characteristic in which cells stop growing because of mutual contact. It is an important regulatory mechanism that maintains the normal morphology of body tissues and prevents disordered cell proliferation in vivo. Contact inhibition allows the cells to stop proliferating when they accumulate in large numbers. Two notable characteristics of many tumor cell lines cultured in vitro are their lack of contact inhibition and the ability to grow without support [42]. When the intercellular density increases, YAP localizes to the cytoplasm, and the activities of cell proliferation-related genes and apoptosis-inhibiting genes are inhibited. In contrast, at a low cell density, YAP localizes to the nucleus and its proliferation-inducing activity is increased; that is, contact inhibition between cells occurs and is mediated by YAP [43]. The integrated membrane protein angiomotin (AMOT) and related proteins can inhibit YAP activity by chelating and binding with YAP in the cytoplasm, thereby inhibiting cell proliferation and intercellular contact and maintaining normal cell density [44]. Zhang et al. [45] found that at high cell densities, overexpression of 14-3-3ζ (an apoptosis inhibitor protein) in human umbilical cord mesenchymal stem cell-exosomes (hucMSC-EX) promoted the binding of LATS to YAP, enhanced the phosphorylation of YAP at ser127, and inhibited YAP activity. After transfecting YAP into the non-neoplastic breast epithelial cell line MCF10A by retrovirus, Overholtzer et al. [46] found that overexpression of YAP could abolish the cell contact inhibition effect, promote growth-factor-independent proliferation, inhibit apoptosis, promote epithelial mesenchymal transition (EMT), and anchor independent growth in soft agar. Taken together, these changes are indicative of malignant transformation of cells.

### 3.4. YAP can Regulate Angiogenesis

The main pathological feature of psoriasis is the painful expansion of superficial dermal vessels with a clinical manifestation, such as the Auspitz sign [47]. Tumor tissues require a nutrient supply, which is often accompanied by extensive angiogenesis [48]. Choi et al. [49] demonstrated that YAP is an important regulator of angiogenesis in mice; after phosphorylation, YAP is inactivated and then redistributed in a cell-contact-dependent manner through E-cadherin. In mice, YAP knockout is associated with a significant reduction in the number of endothelial buds in the common duct network and aortic ring. During angiogenesis, the vascular endothelial growth factor (VEGF)–VEGF receptor 2 (VEGFR2) signaling axis depends on YAP/TAZ activation [50]. VEGFR regulates YAP/TAZ through the Rho GTPase, mitogen-activated protein kinase (MAPK), and phosphatidylinositol 3-kinase (PI3K) pathways to regulate angiogenesis [51,52,53,54]. During postnatal development of human umbilical vein endothelial cells (HUVECs) and mouse retina, Ang2 is a key YAP target gene in endothelial cells that mediates the regulation of angiogenesis and vascular remodeling by YAP [55].

## 4. Research on the Role of YAP in Psoriasis and Skin Tumors

### 4.1. Psoriasis

The pathology of psoriasis is manifested by excessive proliferation and abnormal differentiation of epidermal keratinocytes and dilation and proliferation of superficial dermal vessels; these manifestations are similar to the pathology of skin tumors. Consequently, interest has grown in examining the tumor-related factors and signaling pathways in psoriasis. Jia et al. [56] showed that YAP expression increased in clinical samples of psoriasis and skin lesions in an imiquimod psoriasis mouse model. Cytological experiments showed that following YAP knockout, the proliferation of the human immortalized keratinocyte line HaCaT was inhibited; the cell cycle was blocked in the G0/G1 phase; the apoptosis rate increased; and the expression of cyclins A, B1, D1, and E decreased; whereas the expression of apoptosis-related protein cleaved-caspase-3 increased, which may be related to the change in amphiregulin (AREG) expression. In psoriatic cells and animal models, overexpression of RAS-association domain family 1A (RASSF1A) or inhibition of promoter methylation with the methylation inhibitor 5-Aza-CdR reduced YAP expression; inhibited cell proliferation; induced G0/G1 cell cycle arrest; increased the apoptosis rate; decreased the expression of inflammatory cytokines; decreased the activity of the AKT, ERK, STAT3, and NF-κB signaling pathways; and improved skin lesions in mice, indicating that RASSF1A may be an upstream negative regulator of YAP in psoriasis [57]. Danshensu, a *Salvia miltiorrhiza* extract, was found to inhibit YAP expression in psoriatic cells and animal models and inhibited cell proliferation, caused cell cycle arrest in the G0/G1 phase, promoted cell apoptosis, and improved mouse skin lesions in a concentration-dependent manner [58]. Taken together, the above findings provide a theoretical basis for the use of YAP-targeted drugs in the treatment of psoriasis.

### 4.2. Basal Cell Carcinoma

BCC, the most common malignant skin tumor, occurs on the head and neck in approximately 80% of patients. Although metastasis is infrequent in BCC, local infiltration and tissue destruction often occur, affecting the appearance and quality of life of patients [11]. YAP and its downstream target factors CCN1 (CYR61) and CCN2 (CTGF) are highly expressed in BCC cells. CCN1 regulates the growth and survival of keratinocytes, and CCN2 affects the ECM microenvironment. YAP is primarily expressed in the cytoplasm and nucleus of BCC cells. YAP knockout and restoration of CCN1 expression in human epidermal cells can restore cell proliferation and survival. It has been speculated that CCN1 and CCN2 may play an important role in YAP regulation of cell proliferation and survival [59]. RNA-seq analysis also showed that YAP target genes were overexpressed in BCC tumor samples, indicating that the Hippo-YAP signaling pathway may be involved in the pathogenesis of BCC [60].

### 4.3. Cutaneous Squamous Cell Carcinoma

Basal cSCC is the second most common NMSC, and its malignancy rate is higher than that of BCC. Particularly in patients with chronic immunosuppression, a history of ultraviolet or radiotherapy, and chronic skin injuries (such as wounds, ulcers, or burns), the risk of invasion or metastasis increases [61]. Therefore, it is important to study the pathogenesis, prevention, diagnosis, and treatment strategies of cSCC. In one study, researchers demonstrated that YAP expression gradually increased in normal skin tissue, precancerous lesions, actinic keratosis, and carcinoma in situ Bowen’s disease until developing into cSCC. In vitro cell experiments confirmed that YAP knockdown in the cSCC cell line A431 inhibited cell proliferation, blocked the cell cycle in the G0/G1 phase, promoted apoptosis, and weakened the migration and invasion abilities of tumor cells. The results of in vivo experiments confirmed that YAP knockdown inhibited tumor growth in a transplanted tumor model in nude mice. As a result of YAP overexpression in A431 cells, tumor cells proliferated, the proportion of cells in the S phase increased, and apoptosis by 5-FU was hindered; however, there were no significant effects on migration and invasion. These effects may be mediated through the AREG/Ras/AKT/ERK signaling pathway [62]. S100A7 is highly expressed in psoriasis and cSCC. In the cSCC cell line A431, increased expression of S100A7 was accompanied by the phosphorylation and inactivation of YAP, both of which are regulated by the cell morphology and cell density. The induction of S100A7 was inhibited by the nuclear localization of YAP, which was confirmed by the activation or inhibition of the Hippo signaling pathway via the loss- and/or gain-of-function of LATS1 and MST1. YAP inhibits S100A7 expression by binding to TEAD. Although YAP expression is negatively correlated with S100A7 protein expression, researchers believe that both proteins can promote cell proliferation and inhibit cell differentiation. It has been speculated that S100A7 may function as a replacement for YAP, acting to maintain cell survival and inhibit cell death [63].

### 4.4. Melanoma

YAP is expressed in melanocytes and in melanoma cell lines [64]. The overall survival rate of melanoma patients overexpressing YAP is reduced [65]. Overexpression of YAP in human melanoma cells can enhance the clonal growth and invasion ability of tumor cells, whereas knocking out YAP and TAZ proteins in human melanoma cells can inhibit tumorigenesis, reduce their invasion ability, and inhibit lung metastasis of melanoma in mice [66]. The role of YAP in melanoma may be mediated by stimulation of the LRP1 promoter [67]. In addition, YAP has been found to play an important role in Gq/11-induced tumorigenesis, and YAP can be used as a potential drug target for uveal melanoma (UM) with GNAQ or GNA117 mutations [68]. YAP activation is also associated with resistance to MAPK inhibitors in clinical melanoma therapy [69]. However, Moroishi et al. recently reported that the Hippo pathway suppresses antitumor immune responses in a melanoma model. Furthermore, LATS1/2 ablation improves tumor immunogenicity and suppresses tumor growth both in vivo and in vitro [70].

### 4.5. Other Skin Tumors and Tumor-Related Conditions

Pilomatrixoma, a benign tumor, is derived from primitive epithelial germ cells that differentiate into hairy stromal cells. Cappellesso et al. [71] studied YAP expression in tumor tissues through immunohistochemical analysis of samples from 11 patients with pilomatrixoma. The results showed that the expression of YAP in a large (6.5 cm) pilomatrixoma was higher than that in 10 normal pilomatrixoma samples and was positively correlated with the proliferation rate. YAP was mainly localized to the nucleus of the large pilomatrixoma, whereas YAP was localized in both the nucleus and cytoplasm of normal pilomatrixoma samples. These results indicate that YAP is involved in the growth and proliferation of pilomatrixomas.

Long-term exposure to arsenic leads to skin dryness, melanosis, hyperplasia, and hyperkeratosis and may eventually develop into BCC or cSCC. Arsenic treatment of mouse skin can activate LATS1 kinase and other Hippo signaling regulatory proteins such as Sav1 and MOB1. Phosphorylated LATS1 kinase can inactivate YAP, but this does not occur in the arsenic-treated epidermis. The expression of p-YAP in the epidermis was negatively correlated with the arsenic dose, and the expression of β-catenin was positively correlated with the arsenic dose. Non-phosphorylated YAP was translocated to the nucleus and combined with TEADs to upregulate the expression of downstream genes, such as Cy61, Gli2, Ankrdl, and Ctgf. This may be the underlying mechanism of arsenic-induced skin keratosis [72].

## 5. Conclusions

High levels of YAP expression have been found in various skin diseases—psoriasis, cSCC, BCC, and melanoma—that are often characterized by abnormal cell proliferation and inhibition of apoptosis. The high expression of YAP in these skin diseases may be related to its function and may play an important role in disease pathogenesis. The specific mechanism of the Hippo–YAP signaling pathway in this complex network requires further study. The study of new drugs targeting YAP can provide a novel approach for the treatment of skin diseases [73]. However, there are no ongoing preclinical or clinical trials [74]; this may be because YAP has a wide range of physiological functions, and inhibition of its expression may cause side effects. Therefore, a deeper understanding of the role of the Hippo–YAP signaling pathway in the pathogenesis of these skin diseases is required to uncover more specific targets. Only in this way can we make meaningful progress in the development of drugs for the treatment of these skin diseases.

## Figures and Tables

**Figure 1 jpm-12-00978-f001:**
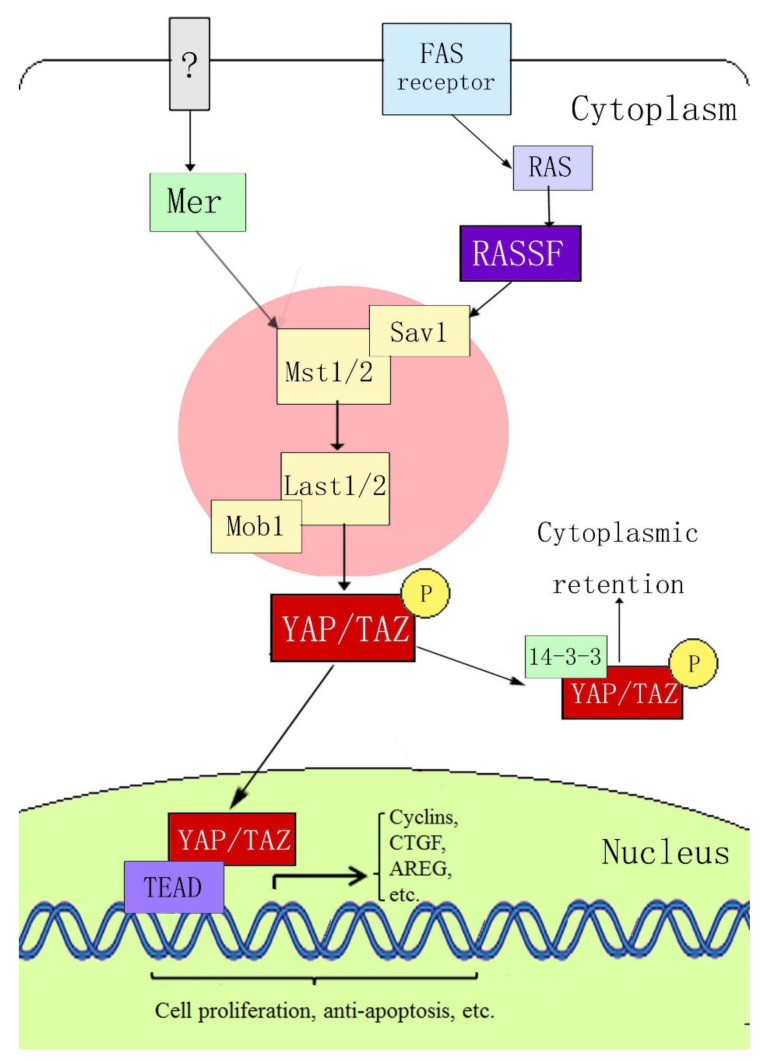
Hippo–YAP signaling pathway.

## Data Availability

Not applicable.

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
