# Peer review of "Role of Yes-Associated Protein in Psoriasis and Skin Tumor Pathogenesis"

_jpm, 2022, doi:10.3390/jpm12060978_

Round 1

Reviewer 1 Report

The article ‘Role of Yes-associated protein in psoriasis and skin tumor pathogenesis’ may be an useful contribution to the journal; however, few changes should be taken into consideration:

Line 34 please reconsider: ‘Psoriasis is a chronic inflammatory skin disease that occurs in young adults’ – this phrase is limitting, as psoriasis affects children and elderly individuals, as well.

Lines 45-47: Although biological agents have achieved good results in treating psoriasis in recent years, because of the high cost and the elimination of infectious diseases such as tuberculosis, hepatitis, and tumor diseases, there remain many limitations, and  recurrence cannot be avoided. – this needs to be rephrased, as the current form of the phrase is a bit tedious and prolix.

Paragraph containing lines 50-65 needs re-organization, as to be clear to the reared that there are melanoma, on one side, and NMSC on the other; the current form creates unwanted complexity, instead of organizing and simplifying the matter.

Incidence rates (approximately) are necessary in the Introduction, for both psoriasis and melanoma/NMSCs, as to inform the reader about the magnitude of the problem.

Line 66- there are many known factors in the etiology of skin cancers (chemicals, UV radiation, exposure to X-rays, to name a few); this should not be oversimplified. In the same time, psiroasis etiology remains in shadow, at least partially. Please rephrase the paragraph as to relate to current state-of-the-art knowledge on the matter

Methodology is not clear; authors should state how the extraction of the articles was performed, in a separate section, in order to ensure reproductibility and to minimise selection bias.

Conclusion section is too long, and also introduces new information; the first part of the paragraph should be moved to Introduction or elsewhere.

Also, there is a need to clarify the need to join the two otherwise different pathologies, psoriasis (an inflammatory disorder) and cutaneous tumors, in the same article. Possible simmilarities and discrepancies, as well as potential reasons for the same pathway involvement in the 2 distinct type of conditions should be taken into consideration and discussed, in the best interest of the reader.

Grammar and punctuation must also be carefully checked within the entire article.

Author Response

We thank you for your thoughtful suggestions. The manuscript has benefited from these insightful suggestions. The manuscript has been rechecked and the necessary changes (highlighted in yellow) have been made in accordance with your suggestions. The responses to all comments have been prepared and attached below.

The article ‘Role of Yes-associated protein in psoriasis and skin tumor pathogenesis’ may be an useful contribution to the journal; however, few changes should be taken into consideration:

Line 34 please reconsider: ‘Psoriasis is a chronic inflammatory skin disease that occurs in young adults’ – this phrase is limitting, as psoriasis affects children and elderly individuals, as well.

We thank the reviewer for the constructive suggestion and revised this sentence.

Lines 45-47: Although biological agents have achieved good results in treating psoriasis in recent years, because of the high cost and the elimination of infectious diseases such as tuberculosis, hepatitis, and tumor diseases, there remain many limitations, and  recurrence cannot be avoided. – this needs to be rephrased, as the current form of the phrase is a bit tedious and prolix.

As suggested by the reviewer, we have rephrased this sentense.

Paragraph containing lines 50-65 needs re-organization, as to be clear to the reared that there are melanoma, on one side, and NMSC on the other; the current form creates unwanted complexity, instead of organizing and simplifying the matter.

As suggested by the reviewer, we have rephrased this sentense.

Incidence rates (approximately) are necessary in the Introduction, for both psoriasis and melanoma/NMSCs, as to inform the reader about the magnitude of the problem.

We thank the reviewer for the constructive suggestion and revised the manuscript accordingly.

Line 66- there are many known factors in the etiology of skin cancers (chemicals, UV radiation, exposure to X-rays, to name a few); this should not be oversimplified. In the same time, psiroasis etiology remains in shadow, at least partially. Please rephrase the paragraph as to relate to current state-of-the-art knowledge on the matter

We thank the reviewer for the constructive suggestion and revised the manuscript according to your suggestion.

Methodology is not clear; authors should state how the extraction of the articles was performed, in a separate section, in order to ensure reproductibility and to minimise selection bias.

All the refereces in the manuscript were searched and acquired from the Pubmed, summarizd by ourselves. Because it is just a narrative review so we do not list a separate Methods paragraph but explained it in Introduction.

Conclusion section is too long, and also introduces new information; the first part of the paragraph should be moved to Introduction or elsewhere.

We have shortened this paragraph and moved some parts of it to sction 3.

Also, there is a need to clarify the need to join the two otherwise different pathologies, psoriasis (an inflammatory disorder) and cutaneous tumors, in the same article. Possible simmilarities and discrepancies, as well as potential reasons for the same pathway involvement in the 2 distinct type of conditions should be taken into consideration and discussed, in the best interest of the reader.

We have discussed their similarities in pathogenesis in the manuscript.

Grammar and punctuation must also be carefully checked within the entire article.

Thank you and we have checked the grammar and punctuation carefully.

Reviewer 2 Report

None

Author Response

We thank you for your recognization for our manuscipt. 

Reviewer 3 Report

I would like to congratulate the authors of the article entitled "Role of Yes-associated protein in psoriasis and skin tumor pathogenesis". 

This review addresses YAP-associated disease mechanisms and the latest research on psoriasis and various skin tumors. 

The review is well structured, clearly presented, and well documented using recent references.

I would like to suggest that in introduction lines 37-38 "Psoriasis is a persistent skin disease that endangers human physical and mental health" it should be noted that psoriasis is a chronic disease rather than persistent.

Regarding "endangers human physical and mental health" - I think it would be good to list the comorbidities of psoriasis. (DOI: 10.3390/nu13010163 / DOI: 10.1001/jamadermatol.2022.1081 )

"Although biological agents have achieved good results in treating psoriasis in recent years, because of the high cost and the elimination of infectious diseases such as tuberculosis, hepatitis, and tumor diseases, there remain many limitations, and recurrence cannot be avoided" - Is it not clear to me whether the elimination of infectious diseases leads to recurrences?

    •  

    Author Response

    We thank you for your thoughtful suggestions. The manuscript has benefited from these insightful suggestions. The manuscript has been rechecked and the necessary changes (highlighted in yellow) have been made in accordance with your suggestions. The responses to all comments have been prepared and attached below.

    I would like to congratulate the authors of the article entitled "Role of Yes-associated protein in psoriasis and skin tumor pathogenesis". 

    This review addresses YAP-associated disease mechanisms and the latest research on psoriasis and various skin tumors. 

    The review is well structured, clearly presented, and well documented using recent references.

    I would like to suggest that in introduction lines 37-38 "Psoriasis is a persistent skin disease that endangers human physical and mental health" it should be noted that psoriasis is a chronic disease rather than persistent.

    We thank the reviewer for the suggestion and revised the manuscript accordingly.

    Regarding "endangers human physical and mental health" - I think it would be good to list the comorbidities of psoriasis. (DOI: 10.3390/nu13010163 / DOI: 10.1001/jamadermatol.2022.1081 )

    We have added the discription of comorbidities of psoriasis in the revised manuscipt.

    "Although biological agents have achieved good results in treating psoriasis in recent years, because of the high cost and the elimination of infectious diseases such as tuberculosis, hepatitis, and tumor diseases, there remain many limitations, and recurrence cannot be avoided" - Is it not clear to me whether the elimination of infectious diseases leads to recurrences?

    We have rephrased this sentences in the revised manuscript.

    Reviewer 4 Report

    The manuscript titled "Role of Yes-associated protein in psoriasis and skin tumor pathogenesis" is generally speaking a well-organized review into the role of YAP signaling pathway in two prevalent skin diseases, psoriasis and skin tumor, which share certain features in their programs. The authors systematically introduced the molecular composition of YAP signaling pathways, the function of YAP in the skin malignancies and a brief summary of the specific research progress on psoriasis and skin tumors. However, authors should pay more attention to the language and pictures they use because bad wordings and poor picture representation of signaling pathways undermines the communication values of the manuscript and should be addressed. With a better signaling pathway picture, and preferentially including a picture representation of recent research summaries on the YAP and psoriasis/skin tumors, many more readers may find this manuscript appealing and eye-opening.

    1. Abstract Line 26 "High expression of YAP in psoriasis and skin tumors may be related to its function" change to "high.....tumors may indicate their positive functions in the skin inflammation and malignancies"

    2. Line 36. "putting great psy..." this sentence prompts me to believe authors should find someone to polish the language.

    3. Line 43. Never a hyphen before vowel.

    4. Line 53. Metastatic tumors should not be aligned with other types of tumors.

    5. Why is there no chapter 3 but directly jump to 3.1 at Line 94?

    6. Figure 1 is indeed a poorly made. Make a new one using your own art work.

    7. YAP's role in apoptosis is bi-directional and context-dependent, may authors elaborate the reasons? Why is it context dependent? Any literature?

    8. Line 200 "indicating that it may be an ..." What does "it" represent here?

    9. Line 258. Authors may wanted to introduce the animal model this research used to show LATS1/2 ablation led to the suppression of tumor growth.

    Author Response

    We thank you for your thoughtful suggestions. The manuscript has benefited from these insightful suggestions. The manuscript has been rechecked and the necessary changes (highlighted in yellow) have been made in accordance with your suggestions. The responses to all comments have been prepared and attached below.

    The manuscript titled "Role of Yes-associated protein in psoriasis and skin tumor pathogenesis" is generally speaking a well-organized review into the role of YAP signaling pathway in two prevalent skin diseases, psoriasis and skin tumor, which share certain features in their programs. The authors systematically introduced the molecular composition of YAP signaling pathways, the function of YAP in the skin malignancies and a brief summary of the specific research progress on psoriasis and skin tumors. However, authors should pay more attention to the language and pictures they use because bad wordings and poor picture representation of signaling pathways undermines the communication values of the manuscript and should be addressed. With a better signaling pathway picture, and preferentially including a picture representation of recent research summaries on the YAP and psoriasis/skin tumors, many more readers may find this manuscript appealing and eye-opening.

    1. Abstract Line 26 "High expression of YAP in psoriasis and skin tumors may be related to its function" change to "high.....tumors may indicate their positive functions in the skin inflammation and malignancies"

    We thank the reviewer for the constructive suggestion and revised the manuscript accordingly.

    1. Line 36. "putting great psy..." this sentence prompts me to believe authors should find someone to polish the language.

    We thank the reviewer for pointing this out and we have already polished the language by Editage, an authorized Language Editing survice. The certificate was also attached.

    1. Line 43. Never a hyphen before vowel.

    I am sorry but we did not find the specific instance that the Reviewer had flagged. I did find a few other instances of a hyphen before a vowel.

    e.g., “yes-associated protein”,  “apoptosis-inhibiting genes”

    In each case the hyphen precedes a word that begins with a vowel. But they are all fixed nouns that cannot be changed. We have searched several academic style guides (including the Chicago Manual of Style) for such a rule; there doesn’t appear to be one. So we haven't made any changes to this point. If you still think there is a problem, please tell us in detail where we need to modify.

    1. Line 53. Metastatic tumors should not be aligned with other types of tumors.

    We have rephrased this sentences in the revised manuscript.

    1. Why is there no chapter 3 but directly jump to 3.1 at Line 94?

    We are sorry for this mistake and revised it in the revised manuscript.

    1. Figure 1 is indeed a poorly made. Make a new one using your own art work.

    We are sorry that Fig.1 are not satisfied for you but it is indeed made by ourselves. Although it is not very gorgeous, we think it has clearly explained the Hippo pathway.

    1. YAP's role in apoptosis is bi-directional and context-dependent, may authors elaborate the reasons? Why is it context dependent? Any literature?

    Because YAP can both inhibit and promote apoptosis in different cells and conditions. We have illustrated this in Section 3.1.

    1. Line 200 "indicating that it may be an ..." What does "it" represent here?

    It indicated RASSF1A and we have revised it in the revised manuscript.

    1. Line 258. Authors may wanted to introduce the animal model this research used to show LATS1/2 ablation led to the suppression of tumor growth.

    LATS1/2 ablation led to the suppression of tumor growth both in cells and animals. We have indicated it in the revised manuscrpt.

    Round 2

    Reviewer 1 Report

    The manuscript has been improved significantly; I have no further comments.

    Reviewer 4 Report

    The authors responded to the comments very quickly and returned with a polished form of manuscript, which should be of publication quality.

    For my comment #3: "

    1. Line 43. Never a hyphen before vowel."

    Specifically, "...genetic background and other susceptible factors lead to patholog- 43 ical changes such as excessive proliferation..." There is a hyphen after g, before starting a new line, then ical. According to this website (英文作文换行时,单词怎么添加连字符“-”_百度知道 (baidu.com)), such truncation of syllable should be avoided. That means you could use gi-cal but not g-ical when it involves changing to the second line. It is also of note in the new version pathological was moved away from the end of the line so no hyphen was needed.

    For my comments #6: 

    1. Figure 1 is indeed a poorly made. Make a new one using your own art work.

    We are sorry that Fig.1 are not satisfied for you but it is indeed made by ourselves. Although it is not very gorgeous, we think it has clearly explained the Hippo pathway.

    I am sorry I made a wrong judgement. I just hope you could make a better one next time, there are ample examples of signaling pathway art in publications.